# 3D Printed Poly(𝜀-caprolactone)/Hydroxyapatite Scaffolds for Bone Tissue Engineering: A Comparative Study on a Composite Preparation by Melt Blending or Solvent Casting Techniques and the Influence of Bioceramic Content on Scaffold Properties

**DOI:** 10.3390/ijms23042318

**Published:** 2022-02-19

**Authors:** Sara Biscaia, Mariana V. Branquinho, Rui D. Alvites, Rita Fonseca, Ana Catarina Sousa, Sílvia Santos Pedrosa, Ana R. Caseiro, Fernando Guedes, Tatiana Patrício, Tânia Viana, Artur Mateus, Ana C. Maurício, Nuno Alves

**Affiliations:** 1Centre for Rapid and Sustainable Product Development (CDRSP), Polytechnic Institute of Leiria, 2411-901 Leiria, Portugal; sara.biscaia@ipleiria.pt (S.B.); ana.r.fonseca@ipleiria.pt (R.F.); tatiana.patricio@ipleiria.pt (T.P.); taniaviana@gmail.com (T.V.); artur.mateus@ipleiria.pt (A.M.); nuno.alves@ipleiria.pt (N.A.); 2Veterinary Clinics Department, Abel Salazar Biomedical Sciences Institute (ICBAS), University of Porto (UP), Rua de Jorge Viterbo Ferreira, n 228, 4050-313 Porto, Portugal; m.esteves.vieira@gmail.com (M.V.B.); ruialvites@hotmail.com (R.D.A.); anacatarinasoaressousa@hotmail.com (A.C.S.); s.santospedrosa@gmail.com (S.S.P.); rita.caseiro.santos@gmail.com (A.R.C.); fernando.t.guedes@gmail.com (F.G.); 3Animal Science Studies Centre (CECA), Agroenvironment, Technologies and Sciences Institute (ICETA), University of Porto (UP), Rua D. Manuel II, Apartado 55142, 4051-401 Porto, Portugal; 4CBQF—Centre of Biotechnology and Fine Chemistry—Associated Laboratory, Faculty of Biotechnology, Catholic University of Portugal, Rua de Diogo Botelho 1327, 4169-005 Porto, Portugal; 5Vasco da Gama Research Center (CIVG)/Vasco da Gama University School (EUVG), Av. José R. Sousa Fernandes 197, Lordemão, 3020-210 Coimbra, Portugal; 6Veterinary Clinics Department, Vasco da Gama University School (EUVG), Av. José R. Sousa Fernandes 197, Lordemão, 3020-210 Coimbra, Portugal

**Keywords:** polycaprolactone, hydroxyapatite, composites, solvent casting, melt blending, 3D printing, cytocompatibility, bone tissue engineering

## Abstract

Bone tissue engineering has been developed in the past decades, with the engineering of bone substitutes on the vanguard of this regenerative approach. Polycaprolactone-based scaffolds are fairly applied for bone regeneration, and several composites have been incorporated so as to improve the scaffolds’ mechanical properties and tissue in-growth. In this study, hydroxyapatite is incorporated on polycaprolactone-based scaffolds at two different proportions, 80:20 and 60:40. Scaffolds are produced with two different blending methods, solvent casting and melt blending. The prepared composites are 3D printed through an extrusion-based technique and further investigated with regard to their chemical, thermal, morphological, and mechanical characteristics. In vitro cytocompatibility and osteogenic differentiation was also assessed with human dental pulp stem/stromal cells. The results show the melt-blending-derived scaffolds to present more promising mechanical properties, along with the incorporation of hydroxyapatite. The latter is also related to an increase in osteogenic activity and promotion. Overall, this study suggests polycaprolactone/hydroxyapatite scaffolds to be promising candidates for bone tissue engineering, particularly when produced by the MB method.

## 1. Introduction

Bone tissue engineering allows the development of alternative strategies to repair bone defects caused by disease or trauma that cannot be healed spontaneously in the living tissue. This approach emerged due to the limitations of the conventional therapies that usually includes transplantation, surgical reconstitution, and artificial prosthesis. Bone tissue engineering requires the incorporation of osteoblasts on 3D scaffolds for the adhesion, proliferation and differentiation of cells, as well as nutrient and oxygen availability [1,2,3].

There are several requirements that should be considered to obtain an efficient 3D scaffold. In the initial stage, the biocompatibility and bioactivity of the support have a major impact on cell attachment, and an appropriate macro and micro porosity and shape allow the tissue in-growth and delivery of nutrients and oxygen to the cells. Furthermore, the scaffold should have sufficient mechanical strength to provide the structural requirements of the substituted tissue, and the degradation rate must be gradual for cell growth and proliferation to promote the creation of a new bone tissue [2,4,5].

The fabrication of 3D scaffolds with high resolution and control on micro/nano level structure are facilitated using 3D printing technologies. Extrusion-based techniques consist in a computer-controlled layer-by-layer deposition where polymer filament fused is extruded through a nozzle. This technology has been used to process various biomaterials, mainly thermoplastics, with polycaprolactone (PCL), polylactic acid (PLA), and poly(lactic-co-glycolic) acid (PLGA) being the most frequently used for bone tissue engineering applications [6,7,8,9,10,11,12,13]. However, due to their low bioactivity, the development of composite scaffolds has been especially explored by combining polymers with calcium-phosphate-based materials [14]. The composites are comprised of two or more materials, aiming at the development of more efficient scaffolds by combining regenerative properties of more than one biomaterial [15]. In addition, composites integrating calcium phosphates and polymers combine good mechanical properties with good biocompatibility, reaching to a 3D substitute that mimics the heterogeneity and hierarchical structure typical of the native extracellular bone matrix [16,17]. Therefore, the use of hydroxyapatite (HA) can be justified because of its chemical similarity to the natural mineral component of bone tissue. Additionally, HA has the ability to integrate into bone structures and support its growth without breaking or dissolving the tissue (bioactive properties) [18,19]. HA has been previously blended with PCL, a polymer considered safe by the Food and Drug Administration (FDA) [20]. PCL, through its melt processing, allows 3D porous scaffolds with highly interconnected porous network and with high reproducibility to be obtained by using commercialised [21] and novel biomanufacturing systems [5,8,9,10,11,12,22,23,24]. PCL-based scaffolds have been produced by these newly developed equipments and have been tested in vitro and in vivo with good cell viability, osteogenic differentiation promotion and improved bone tissue regeneration outcomes, both in vitro and in vivo [5,8,11,25,26,27,28,29,30,31].

With respect to the production of the composite structures, their properties are mainly dependent on the nature of the materials and preparation method. In the present study, one of the objectives considered is to investigate how the solvent can influence the resultant composite mixture and its properties. Concerning this topic, a few relevant studies regarding solvents’ impact in the solvent casting technique have been reported. Patlolla et al. [32] conducted a study where 20% HA/80% β-TCP and PCL composites were produced using either methylene chloride (MC) or a combination of MC and dimethylformamide (MC + DMF) solvents. This study demonstrates that the produced structures presented uniform fibres and homogeneous ceramic dispersions, and the solvent or solvent combination used to produce the composites proved to be a determining factor to define its properties, thus affecting cell growth kinetics. In another study, Choudhury and colleagues [33] investigated the effect of different solvents, 1,1,1,3,3,3-hexafluoro-2-propanol (HFIP), dichloromethane (DCM) and chloroform (CF), to fabricate PLA scaffolds. PLA/DCM scaffolds presented more thermal stability and a stiffer base compared with PLA/HFIP and PLA/CF scaffolds. However, the PLA/CF scaffold showed higher porosity against the PLA/HFIP scaffold, which is beneficial for the requirements for bone tissue engineering. 

Altogether, it has been established that different solvents interact with the polymer quite differently. These findings appeared to be caused by (I) proton conductivity [34,35,36], (II) solvent volatility [33,34], which has been found to be better with a less volatile solvent, combined with (III) stronger solvent–polymer interactions [34], to address the uniformity requirements of the scaffold/structure and the regularity of polymer surface. Additionally, the choice of solvent can impact the dispersion of particles within a polymeric matrix, and all these features can affect composite structure, morphology and properties, which may, in turn, affect cell behaviour and tissue in-growth and regeneration [32].

In this work, in addition to the incorporation of a ceramic amount of 20 and 40 wt% on PCL, we analyse the influence of the two mixture techniques (solvent casting (SC) and melt blending (MB)) on the properties of PCL and PCL/HA composites. Following this, the six different groups considered (PCL SC, PCL MB, PCL/HA 80:20 SC, PCL/HA 80:20 MB, PCL/HA 60:40 SC and PCL/HA 60:40 MB) are 3D printed through an extrusion-based technique, using a previously developed biomanufacturing system [8,9,10,12,22,24,25,26,27,28,29], and their physicochemical properties are further analysed. Further in vitro assays are taken, applying mesenchymal stem cells so as to evaluate the regenerative potential of the produced scaffolds. 

## 2. Results

### 2.1. Fabrication and Characterization of 3D Scaffolds

The PCL and PCL/HA composites were used for the fabrication of 3D printed scaffolds. The chemical structure of the PCL and PCL/HA scaffolds was analysed using FTIR-ATR spectroscopy (Figure 1A). The spectra of all fabricated scaffolds present structural bands of PCL and HA that only differ in terms of intensity. The characteristic peaks of PCL appear at 2865 and 2941 cm^−1^ that correspond to the symmetric and asymmetric stretching of the CH_2_ group, and 1720 cm^−1^ related to carbonyl stretching (C=O). Furthermore, the band at 1239 cm^−1^ correspond to the C–O and C–C stretching characteristic of the crystalline phase and at 1164 cm^−1^ is related to the symmetric COC stretching. Regarding HA, the typical bands are present at 1088, 600 and 568 cm^−1^, which are attributed to ʋ_3_ and ʋ_4_ of P–O stretching. 

Figure 1B presents the element analysis of the PCL/HA scaffolds, which also confirms the incorporation of HA in the composites with the presence of phosphorous and calcium elements. 

The thermal properties of PCL and PCL/HA scaffolds were assessed by DSC thermograms, shown in Figure 2, and the main data is reported in Table 1. The crystallization temperatures present a small decreased in PCL/HA 60:40 composites and the incorporation of the ceramic kept the melting temperature of neat PCL. Furthermore, the endothermic melting enthalpies decreased with the addition of HA. The degrees of crystallinity in SC samples, however, are similar with an evident increase in MB composites. 

The thermal stability of the PCL and PCL/HA composites and the evidence of the amount of the HA in the composites were analysed by TGA thermograms and is presented in Figure 3. The thermograms reveal a significant one step weight loss with degradation temperatures between 369 and 387 °C (Table 2), which corresponds to the decomposition of PCL. Above 450 °C, no weight loss was observed. The results present a slight decrease in the degradation temperature with the addition of the bioceramic, as well as a decrease in mass loss, as a function of the amount of HA present in the sample. 

Figure 4A shows an example of a successfully fabricated scaffold with an interconnected porous structure. Further, the filament orientation, pores interconnectivity and porosity of all scaffolds were evaluated by Micro-CT (Figure 4B–D). The results suggest that the fabricated scaffolds present homogeneous 3D structure, as the 3D designed scaffold. Most of the scaffolds have a similar porosity ranging from 58–60%, with the PCL MB scaffold standing out with 64%.

SEM micrographs of PCL, PCL/HA 80:20 and PCL/HA 60:40 scaffolds of the two blend methods (SC and MB) are shown in Figure 5A. As for the qualitative analysis of the images, MB and SC PCL scaffolds show filament surface with a small corrugation that became less visible in the PCL/HA groups. On the other hand, PCL/HA scaffolds present a homogeneous distribution of the HA on the polymer matrix, with some particles exposed on the filament surface. This aspect is more visible with higher amounts of HA (PCL/HA 60:40), where some particles agglomerations are also observed. Additionally, SC scaffolds still show micropores maintaining this appearance with a substantial number of pores, but with a larger size in the PCL/HA (60/40 wt%) formulation. As for the semi-quantitative analysis of the SEM images presented in Figure 5, the 3D printed structures present interconnected pores and uniform pore sizes. The filament and pore size were measured to analyse the structural characteristics of the proposed scaffolds and the results are consistent with those of the design parameters. Between the groups produced by the SC method, the PCL/HA 60:40 and 80:20 groups presented pores with larger sizes, compared to the PCL group. For the six groups, the filament diameters of the scaffolds are mainly in the range of 300–306 µm for a nozzle with an inner diameter of 300 µm, and the pore size is in the range of 300–318 µm, smaller than the 350 µm established in the scaffold design parameters.

The compressive modulus for the different scaffolds is presented in Figure 6 and all results from the mechanical analysis are listed in Table 2. The incorporation of HA in the PCL polymer matrix contributes to the increase in the mechanical properties. However, they do not increase proportionally to the enhancement of the ceramic concentration in the composite. In fact, differences between PCL/HA 80:20 and 60:40 are not statistically significant. Furthermore, the MB method presented for all groups better outcomes regarding mechanical properties, when compared to the SC groups. Finally, the MB method combined with the addition of HA provides the best mechanical response of the scaffolds (compressive modulus and maximum stress).

### 2.2. Cytocompatibility Assays

#### 2.2.1. Cell Viability Assay

A preliminary Presto Blue^TM^ viability assay was conducted with human dental pulp stem/stromal cells (hDPSCs) on the PCL MB and SC scaffolds, to assess cellular viability and thus, the scaffolds’ cytocompatibility. A control group with cells seeded directly on the well, without scaffold, was considered. Figure 7 and Table 3 present the results obtained. The hDPSCs cell population was selected for this assay, as they have shown promising results towards osteogenic differentiation and potential for bone regeneration in previous works [37,38,39], thus being an appropriate choice for this work’s purpose. The results suggest that both PCL scaffolds induce comparable cell viability outcomes, although with the MB group presenting slightly better results over the entire duration of the assay. The population’s health and normal behaviour in culture was also confirmed by the control group. Considering the results obtained on the preliminary cytocompatibility assessment, both the PCL MB and the PCL SC group could be considered as control groups for further studies. However, the PCL MB group presented slightly better outcomes, in agreement with the biomaterials characterization results of both scaffolds. From this point forward, PCL MB scaffold was used as a scaffold control group for further in vitro studies where different PCL/HA formulations were evaluated.

A Presto Blue^TM^ viability assay was conducted with hDPSCs on the PCL MB (control) and different PCL/HA scaffolds. A control of the cellular populations’ health and normal behaviour in culture was considered, as described above. Figure 8 and Table 4 and Table 5 present the results obtained for the corrected absorbance and the % of viability inhibition, when normalized to the values to the control group (PCL MB). The results suggest the incorporation of HA to positively influence hDPSCs viability, with the increase in HA content presenting superior outcomes. As for the best blending techniques, differences between the respective SC and MB groups are not statistically significant in this assay, with both MB and SC groups presenting positive hDPSCs viability outcomes.

#### 2.2.2. Osteogenic Differentiation Assay

The osteogenic potential of the scaffolds was assessed by Alizarin Red S (ARS) protocol, following a 21 days incubation period, by the detection of mineral deposition, as described in previous works [40,41]. ARS was extracted and further quantified at 405 nm. The results are presented in Figure 9 and Table 6 and Table 7, suggesting that HA incorporation promotes the osteogenic differentiation of hDPSCs. This promotion is enhanced in the groups with higher HA content. Differences between the respective blending techniques groups (MB and SC) are not statistically significant, similarly to the cell viability assay.

### 2.3. Scanning Electronic Microscopy (SEM)

A SEM analysis was conducted, following in vitro viability assessment. Unseeded scaffolds were also analysed. Images are presented with different magnification in Figure 10. SEM analysis confirms hDPSCs attachment and proliferation on the scaffolds, presenting normal structure and morphology. Differences between groups could not be assessed, and qualitatively all presented positive outcomes regarding 3D cell attachment, cell adhesion and morphological structure.

## 3. Discussion

Three-dimensional porous biodegradable scaffolds were explored using various techniques in the interest of being suitable as bone substitutes for bone repair and reconstruction. The research on the production processes for the dispersion of nanomaterials in polymer matrices and the design of 3D printing scaffolds, as well the combination of these features, plays a critical role in tissue engineering. These processes contribute to achieve the requirements of 3D scaffolds and the appropriate physical inter-connections for an efficient cell permeation and colonization. The current work is focused on the processing of PCL and PCL/HA composites prepared by MB and SC methods, using an extrusion-based technique for the development of 3D substitutes for bone tissue engineering. PCL was selected as the major component of scaffolds because it is easy to process by 3D printing, is biodegradable and biocompatible, and possesses adequate mechanical strength. However, the hydrophobic surface and lack of osteoconductivity of the PCL matrix represents some disadvantages, which can impair cells adhesion and proliferation [42,43]. Therefore, HA incorporation was considered to enhance scaffolds’ performance, providing more strength and improving cellular activities, including cell attachment, proliferation, and differentiation [17,44]. Moreover, in human body nearly 70% of hard tissues are composed of HA and are clinically used for orthopaedic and dental repairs [45]. Considering that usually the amount of ceramic remains under 40% of the final material weight, and the most used polymer/ceramic weight ratio is 80/20, the weight ratio of polymer/ceramic was defined at 80/20 and 60/40 wt% [14]. These proportions allow to evaluate and correlate this work outcomes with other previous works and also to prevent 3D printing difficulties, such as nozzle clogging and nonuniform deposition of the composite material filaments. Regarding the mixture techniques to produce PCL and PCL/HA composites, MB and SC methods are widely used for this purpose, as they allow an easy way to produce composites and provide good dispersion of fillers into the polymer matrix. While the MB method depends on high temperatures to mix the two materials, the SC implies the use of an organic solvent to promote polymer dissolution and improve HA nanoparticles dispersion. The macroporosity of the scaffolds was designed using an extrusion-based 3D printing technique to obtain well-defined and interconnected pores for efficient cell colonization and infiltration, and consequently cell adhesion and in-growth. Following previous valuable works on PCL/HA scaffolds with promising outcomes [13,44,46,47,48,49], this work intended to further analyse the impact of different proportions of HA content, as well as the application of two different fabrication processes for the incorporation of HA nanoparticles in PCL matrices, on the mechanical, physical, chemical, and in vitro outcomes of the scaffolds. Moreover, it is also important to highlight the aim of achieving controlled, reproducible and well-defined 3D structures fabricated by a previously developed 3D printing system [8,9,10,12,22,24,25,26,27,28,29].

The chemical composition of the composites was analysed by FTIR to investigate the functional groups of the PCL/HA composite scaffolds and any chemical interactions among the components of PCL and HA. The spectra of the PCL/HA composite scaffolds presented all characteristic bands of PCL and HA, and only differed in terms of intensity, confirming the lack of chemical interaction. These results are consistent with those reported by others works [50,51]. Chemical analysis indicates that MB and SC composites have been properly prepared and successfully 3D printed with the integration of the characteristic peaks of the individual components.

Regarding EDX mapping data (Figure 1B), the HA was detected as distributed throughout the filaments of composites produced from both preparation methods. Comparing the EDX spectra of scaffolds obtained by the different mixture techniques, it is evident that, in the MB method, bioceramic particles are less embedded on PCL matrix and consequently more elements of the HA are detected on these samples’ surface. The Ca/P ratios of the scaffolds are also presented in Figure 1B. The results show that the Ca/P ratio is different among the produced scaffolds. The samples produced by MB present a Ca/P ratio closer to the stochiometric HA (1.67) [52]. The authors hypothesise these differences to be related with the blending processes, as well as to the presence of some HA particles exposed on the filament surface. These results also support the higher mechanical properties obtained in the samples produced by MB.

Thermal analysis revealed that the thermal behavior of the scaffolds is influenced by solvent addition and the dispersion characteristics of the ceramic in polymer. The degrees of crystallinity in the SC samples are similar, with an evident increase in the MB composites. The addition of HA promotes a decrease in the endothermic melting enthalpies. Koupaei and Karkhaneh [53] and Pedrosa et al. [54] reported the same behaviour in PCL/HA scaffolds and in PCL/HA membranes, respectively. These results can be explained due to the high crystallinity of HA that may alter the crystalline properties of the polymer and accelerate the nucleation of the PCL chain segments. Furthermore, extrusion-based 3D printing technique also induces oriented crystallisation due to the formation of row nuclei, enhanced by the flow stress applied to the molten polymer [55,56]. Polymer crystallinity determines the mechanical properties of the produced scaffolds and is a crucial point when considering bone tissue engineering applications [53,55]. TGA thermograms reveal thermal stability in all samples at the processing temperatures used for composites preparation and scaffolds fabrication. Therefore, SC and MB are confirmed as viable methods for mixture preparation and the extrusion-based technique maintains the integrity of the composites.

The fabricated scaffolds present a homogeneous 3D structure and similar porosity. Micro-CT does not allow a precise analysis of HA particles as the spatial resolution limit is between 6–30 µm. Nevertheless, with this analysis, it was possible to verify that in both mixture methods no aggregates above this size are formed. For this reason, and to compare the morphological characteristics of the fabricated scaffolds, in particular filament and pore size, additionally studies of the scaffolds’ surface were conducted by SEM. As for the qualitative analysis of the images, SEM analysis revealed a homogeneous distribution of the HA on the polymer matrix, with some particles exposed on the filament surface and some particles agglomerations. Cestari et al. demonstrate that ceramic filler creates a certain roughness on the surface of the material, which could improve cell adhesion [57]. The solvent addition in the SC method promotes microporosity and this surface morphology is also corroborated in other works [58,59,60]. DMF is a high boiling point solvent that evaporates slowly. The micropores in the filament surface can be related with these chemical properties, as well as the high polarity. However, as other authors reported, the cause for this distinct architecture is not evident [59,61]. As for the semi-quantitative analysis of the SEM images, the 3D printed structures present interconnected pores and uniform pore sizes. The filament and pore size were measured to analyse the structural characteristics of the proposed scaffolds and results are consistent with those of the design parameters with pore size ~310 µm. The scaffolds’ 3D structure and pore size have a great effect on cell attachment and proliferation, and it has been reported that pore size above 300 μm improved vascularisation and bone ingrowth [59,62,63,64].

Regarding the results presented in Figure 6, the incorporation of HA in the PCL polymer matrix contributes to the increase in the mechanical properties. However, they do not increase proportionally to the increase in the ceramic proportion of the composite. In fact, differences between PCL/HA 80:20 and 60:40 are not statistically significant. The lack of difference between the HA incorporated groups can be explained due to the stabilization on the compressive modulus of a certain concentration value of HA. Furthermore, the MB method presented for all groups better outcomes regarding mechanical properties, when compared to the SC groups. Finally, the MB method, combined with the addition of HA, provides the best mechanical response of the scaffolds (compressive modulus and maximum stress). Furthermore, according to the Micro-CT and SEM results, the differences in the mechanical properties are not caused by the porosity, since it is very similar for all the produced scaffolds. However, the micropores observed in the surface of the SC scaffolds may justify the decrease in the compressive strength of the scaffolds. The addition of a fraction of nucleating agents, in this case HA, has a positive influence on crystallinity and mechanical properties. Aliotta et al. investigated this correlation using various nucleating agents and a semicrystalline polymer [58]. The evaluated 3D structures present compressive strength and modulus within the same range of human cancellous bone, between 2–12 MPa and 0.01–2 GPa, respectively [15,60,61]. Therefore, the PCL/HA scaffolds present adequate mechanical support to be applied as bone tissue substitutes. Additionally, the mechanical properties of the scaffolds can be adjusted as a function of mixture method and HA content, depending on the characteristics of the bone defect. These analyses indicate the MB method as a promising choice for producing scaffolds envisioning bone tissue engineering, combined with the incorporation of HA. 

Regarding the cytocompatibility assessment, outcomes for cell viability show that this assay can be considered viable, as the cell population presented normal growth and proliferation in culture, considering the hDPSCs control group. The preliminary assay compared the cytocompatibility of PCL MB and PCL SC, both presenting similar results with no statistically significant differences. However, during most of the assay, PCL MB presented slightly better outcomes, in agreement with the previous characterization assays outcomes. Thus, from this point forward, the PCL MB group was employed as a scaffold control group for further in vitro studies where different PCL/HA formulations were evaluated. Furthermore, data were analysed according to manufacturing instructions and results were interpreted following ISO 10993-5:2009 “Biological evaluation of medical devices”—Part 5—“Test for in vitro cytotoxicity” guidelines. The PrestoBlue^TM^ assay was used, as it allows live-cell evaluations [65]. Thus, the same cell population and the same scaffolds can be analysed throughout the duration of the experiment. According to the guidelines (annex C), a viability inhibition superior to 30% is considered a cytotoxic effect. None of the observed groups presented a viability inhibition superior to 30% and, therefore, can be considered cytocompatible. Overall, the PCL/HA scaffolds outperformed the PCL MB scaffolds in terms of cellular viability with higher HA content, presenting increasing viability outcomes. Similar results have been previously reported by other groups [15,50,66,67,68,69,70,71]. Regarding blending techniques, differences between the respective SC and MB groups are not statistically significant, with both MB and SC groups presenting positive hDPSCs viability outcomes. Finally, with this assay, the cytocompatibility of all the scaffolds could be confirmed. As for the osteodifferentiation assay, it demonstrated the effect on the osteogenic extension promoted by the incorporation of HA in the PCL scaffolds. All PCL/HA groups presented superior mineral deposition detection, when compared to the PCL MB group. Furthermore, considering the undifferentiated group, the PCL/HA groups were capable of inducing osteogenesis, thus suggesting the incorporation of HA to induce spontaneous osteogenesis [72]. The differences between PCL/HA groups are not relevant, with very similar results between groups. The PCL alone was also capable of inducing, although to a lesser extent, intrinsic osteogenesis, in the undifferentiated group, as other groups have reported before [73,74]. As for the control group, the direct comparison with the scaffolds group should not be considered, as this group consisted of a 2D culture condition, in contrast with the 3D culture condition from the scaffolds. It has been widely accepted that 3D cultures present superior differentiation ability, when comparing with 2D cultures [75,76]. This control group, similarly, to the previous assay, was considered as a control of the cell population health and normal behaviour in culture. hDPSCs were selected for this assay due to their pre-established potential towards the osteogenic line and consequent aptitude towards bone tissue regeneration [37,38]. A previous work compared the differentiation potential of PCL and PCL/HA scaffolds with hDPCs, human umbilical cord mesenchymal stem cells (MSCs) and human bone marrow derived MSCs, with the hDPSCs presenting superior osteogenic outcomes [66]. Other have been successfully applying hDPSCs for the evaluation of PCL/HA scaffolds for bone tissue regeneration [71,77,78].

## 4. Materials and Methods

### 4.1. Composites Preparation and Scaffolds Production

Melt blending and solvent casting methods were used for the preparation of composites with PCL (commercially available as CAPA 6500 − Mw = 50 000, Perstop Caprolactones (Cheshire, UK)) and HA (Sigma Aldrich (St. Louis, MO, USA)), with particle size < 200 nm. In the MB method, the PCL pellets were heated in a mortar at 100 °C. After 40 min, HA was added to the melted PCL under constant agitation until a homogeneous solution was obtained. The resulting composite was left to dry and following sliced into small spheres (≈1.5 cm in diameter), for later deposition in the extrusion-based equipment. In the SC, the solid components (PCL and HA) were dissolved in N,N Dimethylformamide (DMF, from Merck KGaA^®^, Darmstadt, Germany). The total amount of DMF was calculated as a function of PCL mass, considering 2 mL DMF for each 0.5 g PCL [29]. For a complete dissolution of the PCL into the solutions, an ultrasonic homogenizer (UP200Ht, Hielscher, Ultrasound Technology, Teltow, Germany) was used and the following parameters were applied: 4 cycles with power of 50 W, during 5 min with a break of 2 min. After the complete dissolution of PCL and dispersion of HA in DMF, both solutions were mixed using a magnetic stirrer (500 rpm) for 10 min. Following this, the solution was placed into Petri dishes and left to dry in a fume hood until the complete solvent evaporation was achieved. The obtained samples with ≈2 mm thickness were then sliced into small squares for further use. Table 8 summarizes the composition of the materials prepared under these conditions.

Cylindrical scaffolds of PCL and PCL/HA composites were produced using an additive manufacturing system, named as Biomate [24]. Scaffolds were fabricated using a deposition velocity of 300 mm/min, a screw rotation velocity of 10–20 rpm and a melting temperature of 70–80 °C. The final scaffolds presented 10 mm diameter, 2.5 mm height, pore geometry of 0°/90°, pore size of 0.35 mm and filament diameter of 0.3 mm. 

### 4.2. Characterizaton of 3D Scaffolds

The chemical composition of the samples was analysed by Alpha FT-IR spectrometer (Bruker, Kontich, Belgium) and Opus Software. All tests were performed at room temperature, in a spectral range of 400–4000 cm^−1^, with a resolution of 4 cm^−1^ in a total of 64 scans. The 3D printed scaffolds were thermally characterized by Differential Scanning Calorimetry (DSC) and Thermogravimetric Analysis (TGA) techniques, using the STA 6000 equipment (Perkin Elmer, Norwalk, CT, USA). DSC analyses were performed for the evaluation of crystallization and melting temperatures of the scaffolds. The samples were heated from 30 to 100 °C at a heating rate of 10 °C/min and maintained at 100 °C for 2 min. Then, the samples were cooled from 100 °C to 15 °C and heated from 15 °C to 100 °C, at the same rate, to obtain the crystallization (T_c_) and melting temperatures (T_m_), respectively. The analysis of the mass loss of the sample as a function of temperature was performed by the TGA technique, applying a heating cycle from 15 °C to 700 °C, at a rate of 10 °C/min. The degree of crystallinity, X_c_, of composites was calculated from the areas of the corresponding DSC melting peaks using the following equation [79]:(1)$$X_{c}=\frac{\Delta H_{m}}{\Delta H_{0}\times X_{\mathrm{PCL}}\;}\times 100$$
where X_PCL_ represents the weight fraction of PCL in the composite, ΔH_0_ the heat of fusion of 100% crystalline PCL (139.5 J/g [80,81]), and ΔH_m_ the peak area of the melting range considered. All runs were performed in triplicate with samples of 6–7 mg placed in alumina pans. Empty pans were used as reference. The flow rate of nitrogen was 20 mL/min during all the runs. Micro X-ray computed tomographic analysis was performed using a high resolution 1174 Skyscan system (Bruker, Kontich, Belgium). The cylindrically shaped scaffolds were mounted on the stage within the imaging system and scanned at 800 µA current, 50 kV voltage, with an exposer time of 6000 ms, rotation step of 0.6 degrees, frame averaging of 2 and without filter. Following the image acquisition, images were reconstructed to 2D cross-sections, beam hardening was corrected, and sufficient smoothing was applied to remove the excess of background noise. NRecon software was used for the reconstruction. All the parameters of the scan and reconstruction settings were identical to the ones used for the samples. To distinguish the solid polymeric material from the void regions, a global thresholding procedure was performed. Porosity was obtained, through CTAn software, and all the calculations were performed within a Volume of Interest (VOI). Each scaffold formulation obtained from MB and SC (PCL, PCL/HA 80:20, PCL/HA 60:40) was scanned 3 times. The porosity was calculated by the average of 3 measurements, from each structure (a total of 9 measurements per group). Morphological images were obtained using CTVox software (version 2.4). The surface morphology of all 3D constructs was analysed by using a scanning electron microscope (SEM) (VEGA 3, TESCAN, Kohoutovice, Czech Republic) that was operated at a voltage of 15 kV, after coating the structures with gold-palladium. Image J software v1.43 was used to calculate the mean pore size and filament diameter by measuring at least 5 points. The same samples were also studied by energy dispersive X-ray spectroscopy (EDX) at a voltage of 15 kV to investigate the elemental constituents as well as the atomic percentage of the elements. Compression tests were performed on the PCL and PCL/HA 3D scaffolds to evaluate the effect of the addition of the ceramic on the mechanical properties of the polymer matrix. The cylindrical structures were analysed on an INSTRON 4505 equipment in a dry state at a rate of 1 mm/min. Five specimens were tested for each composition.

### 4.3. Cytocompatibility Analysis

#### 4.3.1. Cell Culture and Maintenance

hDPSCs were acquired from AllCells, LLC (Cat. DP0037F, Lot N° DPSC090411-01) and maintained in MEM α, GlutaMAX™ Supplement, no nucleosides (Gibco, 32561029), supplemented with 10% (*v*/*v*) fetal bovine serum (FBS) (Gibco, A3160802), 100 IU/mL penicillin, 0.1 mg/mL streptomycin (Gibco, 15140122), 2,05 µm/mL amphotericin B (Gibco, 15290026) and 10 mM HEPES buffer solution (Gibco, 15630122). Cells were maintained at 37 °C, 80% humidified atmosphere and 5% CO_2_ environment. A previous work details the characterization of these cells [37]. 

#### 4.3.2. Cell Viability Assay

For the assessment of the samples’ cytocompatibility, a Presto Blue^TM^ viability assay was performed with hDPSCs, as previously described by Alvites et al. [82]. The 3D scaffolds were sterilized by gamma radiation at 25 kGy, by a Red Perspex dosimeter. A dynamic seeding protocol was considered for the association of the biomaterial with the cellular system. Briefly, scaffolds were incubated with hDPSCs at a density of 2 × 10^5^ cells per scaffold, for 8 h, at 37 °C, 5% CO_2_. Following this, the scaffolds were placed on a non-treated 24-well plate and concealed with complete media. A final group without scaffolds and with only cells was considered for validation of the cell population health and normal proliferation in culture. The scaffolds were incubated for 24 h, 72 h, 120 h and 168 h and the cells metabolic activity was evaluated by Presto Blue^TM^ viability assay. This assay is based on a resazurin solution, which is reduced by living and metabolically active cells, resulting in colour changes in the medium that are quantitatively assessed by ultraviolet-visible spectrophotometry. For each group, unseeded wells were considered as blanks. At every time point, the culture medium was replaced by fresh complete medium to each well, with 10% (*v*/*v*) of 10× Presto Blue cell viability reagent (Invitrogen, A13262). Following this, cells were incubated for 1 h at 37 °C, 5% CO_2_. At this point, the supernatant was removed from each well and placed on 96-well plate, and further analysed on a Thermo Scientific^TM^ Multiskan^TM^ FC Microplate Photometer (Thermo Scientific^TM^, 51119000), at 570 nm and 595 nm. The Presto Blue assay allows live-cell assay, and as such, the wells were gently washed with Dulbecco’s phosphate-buffered saline solution (DPBS, Gibco, 14190169) until the complete removal of the dye and fresh standard culture medium added. For this reagent, the excitation wavelength was 570 nm, and emission was 595 nm. For each well, absorbance at 595 nm was subtracted from the value obtained at 570 nm. Corrected absorbance values for the seeded wells were further calculated, by the subtraction of the average of the correspondent blank wells. Triplicate measurements were considered for each well at every time point. Data were further processed and normalized to the mean of the gold standard group (PCL MB), and presented in % of viability inhibition, comparing to the gold standard group.

#### 4.3.3. Osteogenic Differentiation Assay

Samples were evaluated as to their capacity to promote or inhibit osteogenic differentiation of the hDPSCs. Similar to the viability assay, the scaffolds were seeded by a dynamic seeding protocol. After 3 days in culture, standard culture media were removed, and osteogenic differentiation media were added (StemPro^TM^ Osteogenesis Differentiation Kit, A1007201, Gibco^TM^). Control wells for each biomaterial group were left in standard culture media. Media were changed every 3 days for 21 days. An Alizarin Red S (ARS) staining solution (TMS-008-C, Merk-Millipore) was used for the semi-quantitative analysis of the osteogenic differentiation process, as described in previous works [40,41]. Briefly, cells were fixated in 4% formaldehyde (3.7–4% buffered to pH7, 252931.1315, Panreac AppliChem) and stained with 40 mM ARS solution. Cells were incubated for 30 min under gentle agitation. From this point, wells were carefully washed with DPBS until dye was removed from the supernatant. Considering the non-transparent characteristic of the samples, a qualitative assessment could not be performed at this point. Further semi-quantitative analysis was performed by adding a 10% acetic acid solution (ARK2183, Sigma-Aldrich) to the wells. The collection of the cells and mineral deposition was further accessed by scraping of the wells. The content of each well was individually placed on an 85 °C water bath for 10 min, following immediate immersion on ice for 5 min. Samples were further centrifuged, and absorbance values at 405 nm were taken in a Thermo Scientific^TM^ Multiskan^TM^ FC Microplate Photometer. 

#### 4.3.4. Scanning Electronic Microscopy (SEM)

Scaffolds previously used on the cytocompatibility studies were further removed from the plates and fixated for SEM analysis, based on the Utah State University Biological Sample Fixation protocol. In brief, scaffolds were washed with 0.1 M HEPES (Merck^®^, PHG0001) 3 times and further fixated in a 2% glutaraldehyde (Merck^®^, G5882) buffered solution overnight. The fixative was then removed, and scaffolds were washed with HEPES 3 times, 5 min each, under gentle agitation. From this point, samples were subjected to a crescent series of ethanol (50%, 70%, 95% and 99%) for dehydration, 2–3 times for 10 to 15 min each. Following this, scaffolds were soaked in a crescent series of hexamethyldisilazane (HMDS—Alfa Aesar, A15139)-alcohol solution (1:2; 1:1; 2:1) until complete impregnation in a 98% HMDS solution, 3 times for 15 min. Finally, HMDS was removed from the wells and left to evaporate on an air flow chamber overnight. Samples were coated with Au/Pd by sputtering (SPI Module Sputter Coater) and the SEM/EDX exam was performed using a high resolution (Schottky) Environmental Scanning Electron Microscope with X-ray microanalysis and the Electron Backscattered Diffraction analysis was performed in a Quanta 400 FEG ESEM/EDAX Genesis X4M in high vacuum mode.

### 4.4. Statistical Analysis

Statistical analysis was performed with GraphPad Prism, GraphPad Software, La Jolla, California, USA. Results are presented as mean ± standard deviation of the mean (SE). A one-way ANOVA with Tukey’s multi-comparison test was employed for statistical analysis. Differences were considered statistically significant at *p* ≤ 0.05. The results’ significance ia presented with the symbol (*), according to *p* values with one, two, three or four of the symbols (*) corresponding to 0.01 < *p* ≤ 0.05, 0.001 < *p* ≤ 0.01, 0.0001 < *p* ≤ 0.001 and *p* ≤ 0.0001, respectively.

## 5. Conclusions

In this study, PCL/HA composites were prepared by melt blending and solvent casting methods and processed using an extrusion-based technology to obtain 3D scaffolds with well-defined geometry and pore interconnectivity. The incorporation of HA increased the degree of crystallinity and improved the mechanical properties of the scaffolds. This effect was also observed on the MB produced scaffolds, comparing to SC scaffolds, possibly affected by the incorporation of the organic solvent DMF. Furthermore, HA presence was correlated with improved osteogenic activity. As for the HA incorporation proportion, overall, no statistically significant differences were observed, although, morphologically, the 60:40 presented the best homogeneity requirements. Considering these preliminary observations, the PCL/HA MB scaffolds presented overall the best outcomes, regarding their mechanical characterization, as well as the in vitro cytocompatibility and osteogenic-promoting potential. They can, thus, represent fair candidates for bone tissue engineering studies and further in vitro and in vivo studies are envisioned as to reinforce and support these findings.

## Figures and Tables

**Figure 1 ijms-23-02318-f001:**
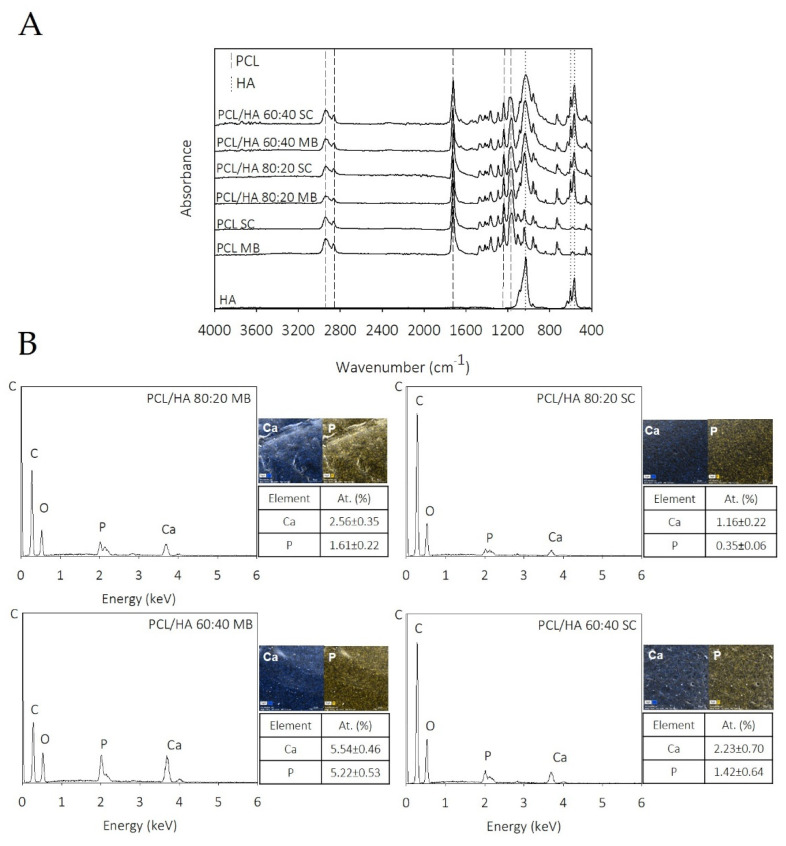
FTIR spectra of pure HA and PCL/HA composites (**A**), and EDX analysis and elemental mapping results showing the elemental composition of PCL/HA composites with predominant calcium and phosphorous atoms (**B**).

**Figure 2 ijms-23-02318-f002:**
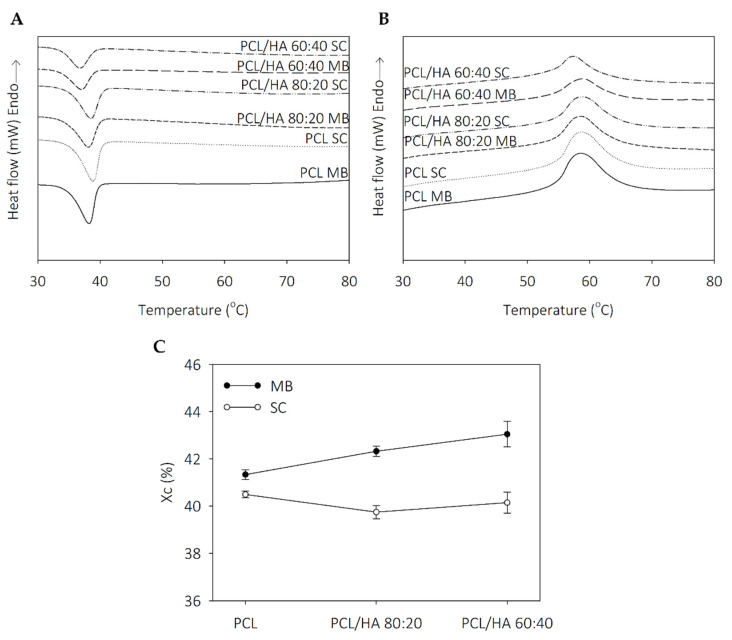
DSC thermograms: (**A**) cooling runs and (**B**) second heating runs; and (**C**) degree of crystallinity of PCL and PCL/HA composites.

**Figure 3 ijms-23-02318-f003:**
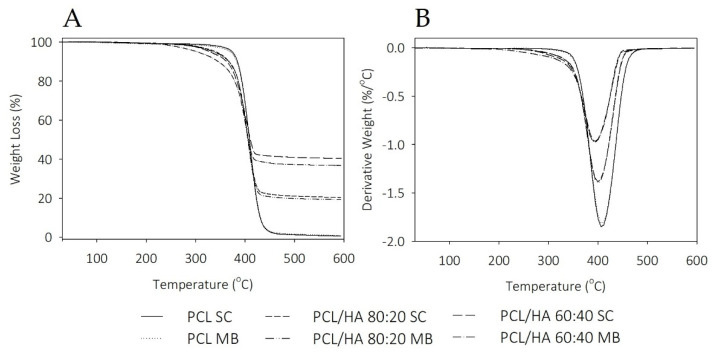
(**A**) TGA and (**B**) DTG results of neat PCL and PCL/HA composites.

**Figure 4 ijms-23-02318-f004:**
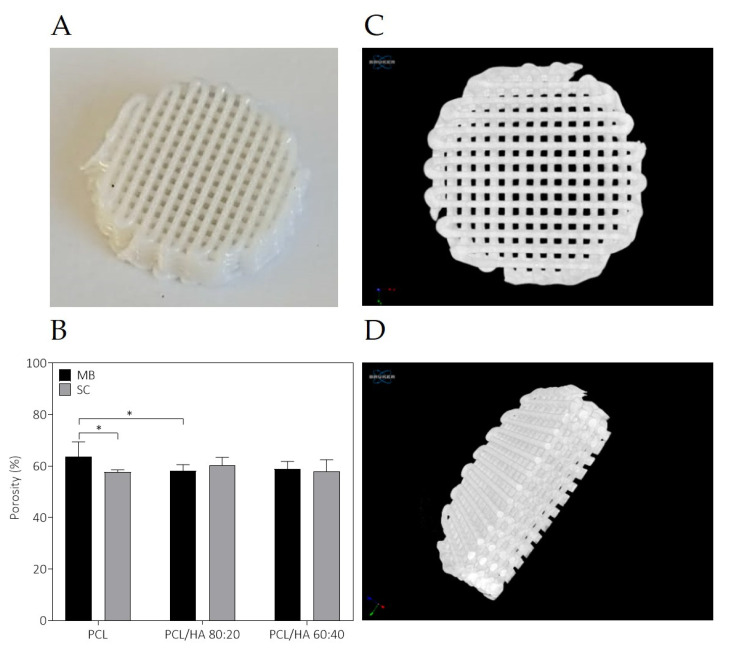
Final appearance of a PCL/HA scaffold (**A**) and porosity obtained of the MB and SC scaffolds (**B**). Micro-CT image of a scaffold allowing the identification of interconnectivity between pores: top-view (**C**) and cross section view (**D**). The results presented in mean ± SE. Differences were considered statistically significant at *p* ≤ 0.05. Results’ significance is presented through the symbol (*), according to the *p* value, with one symbol corresponding to 0.01 < *p* ≤ 0.05.

**Figure 5 ijms-23-02318-f005:**
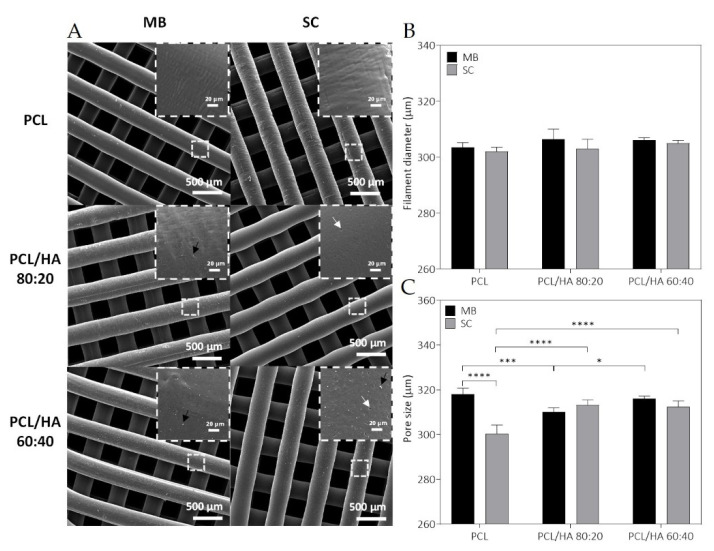
SEM micrographs of PCL and PCL/HA scaffolds, top-view and zoom-in view of the filament (black arrow: HA nanoparticles agglomeration; white arrow: filament micropores) (**A**); the corresponding filament diameter (**B**) and pore size (**C**). The results presented in mean ± SE. Differences were considered statistically significant at *p* ≤ 0.05. Results’ significance is presented through the symbol (*), according to the *p* value, with one, three or four symbols, corresponding to 0.01 < *p* ≤ 0.05; 0.0001 < *p* ≤ 0.001 and *p* ≤ 0.0001, respectively.

**Figure 6 ijms-23-02318-f006:**
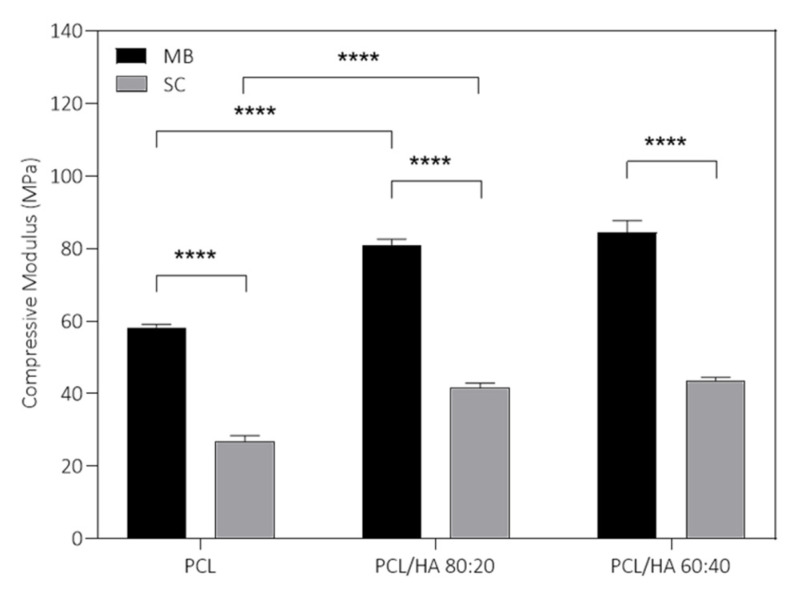
Compressive modulus obtained for the produced scaffolds by MB and SC. The results are presented in mean ± SE. Differences were considered statistically significant at *p* ≤ 0.05. Results’ significance is presented through the symbol (*), according to the *p* value, with four symbols, corresponding to *p* ≤ 0.0001.

**Figure 7 ijms-23-02318-f007:**
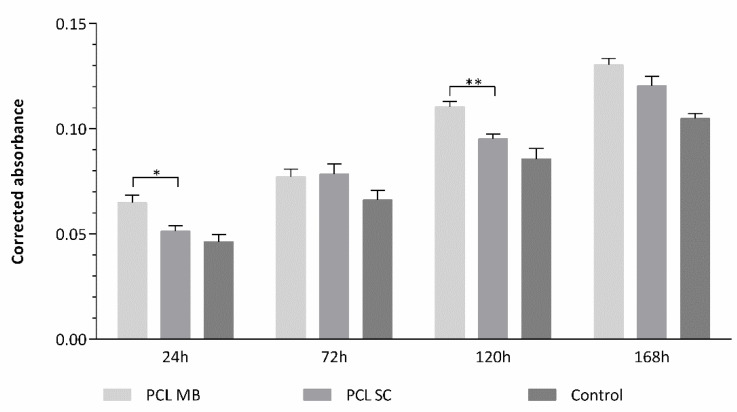
Preliminary cytocompatibility assessment of the PCL groups by Presto Blue^TM^ viability assay for hDPSCs. The results are presented in mean ± SE. Differences were considered statistically significant at *p* ≤ 0.05. Results’ significance is presented through the symbol (*), according to the *p* value, with one or two, corresponding to 0.01 < *p* ≤ 0.05 and 0.001, respectively.

**Figure 8 ijms-23-02318-f008:**
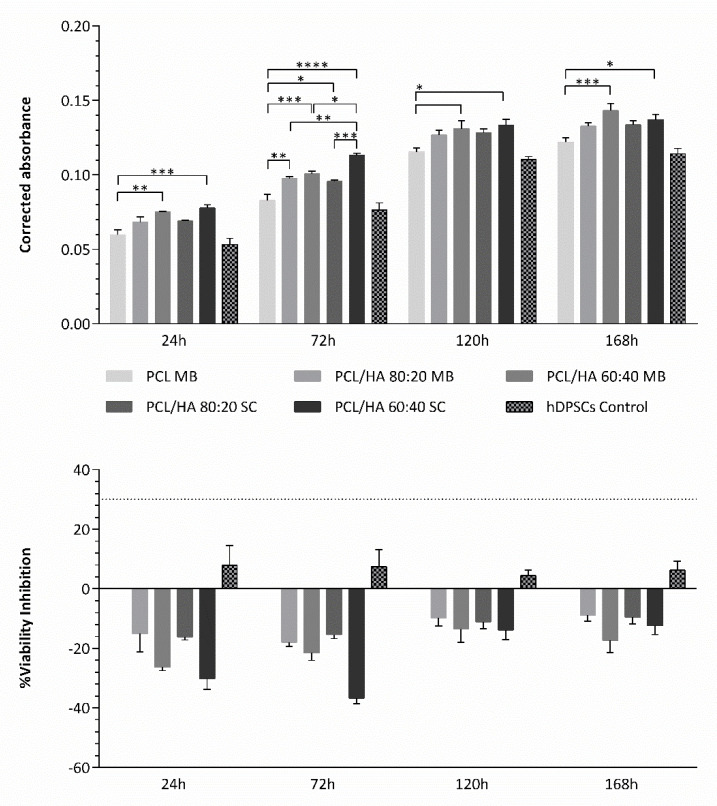
Cytocompatibility assessed by Presto Blue^TM^ viability assay for hDPSCs. The results are presented in mean ± SE. Differences were considered statistically significant at *p* ≤ 0.05. Results’ significance is presented through the symbol (*), according to the *p* value, with one, two, three or four symbols, corresponding to 0.01 < *p* ≤ 0.05; 0.001 < *p* 0.01; 0.0001 < *p* ≤ 0.001 and *p* ≤ 0.0001, respectively.

**Figure 9 ijms-23-02318-f009:**
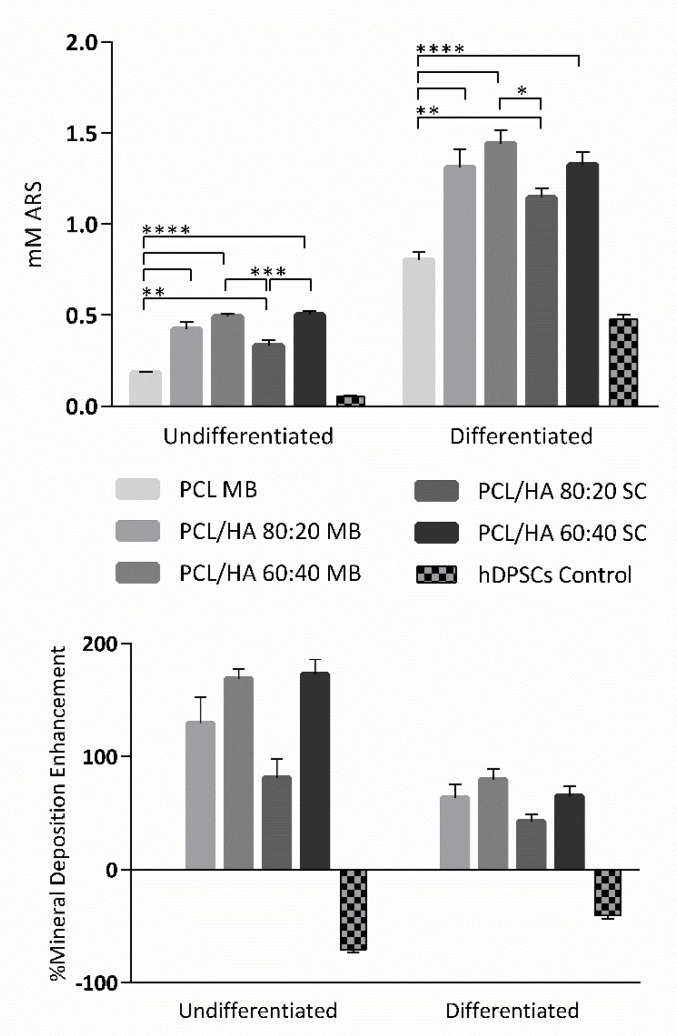
ARS semi-quantification in mM between groups. The results are presented in mean ± SE. Differences were considered statistically significant at *p* ≤ 0.05. The results’ significance is presented through the symbol (*), according to the *p* value, with one, two, three or four symbols, corresponding to 0.01 < *p* ≤ 0.05; 0.001 < *p* 0.01; 0.0001 < *p* ≤ 0.001 and *p* ≤ 0.0001, respectively.

**Figure 10 ijms-23-02318-f010:**
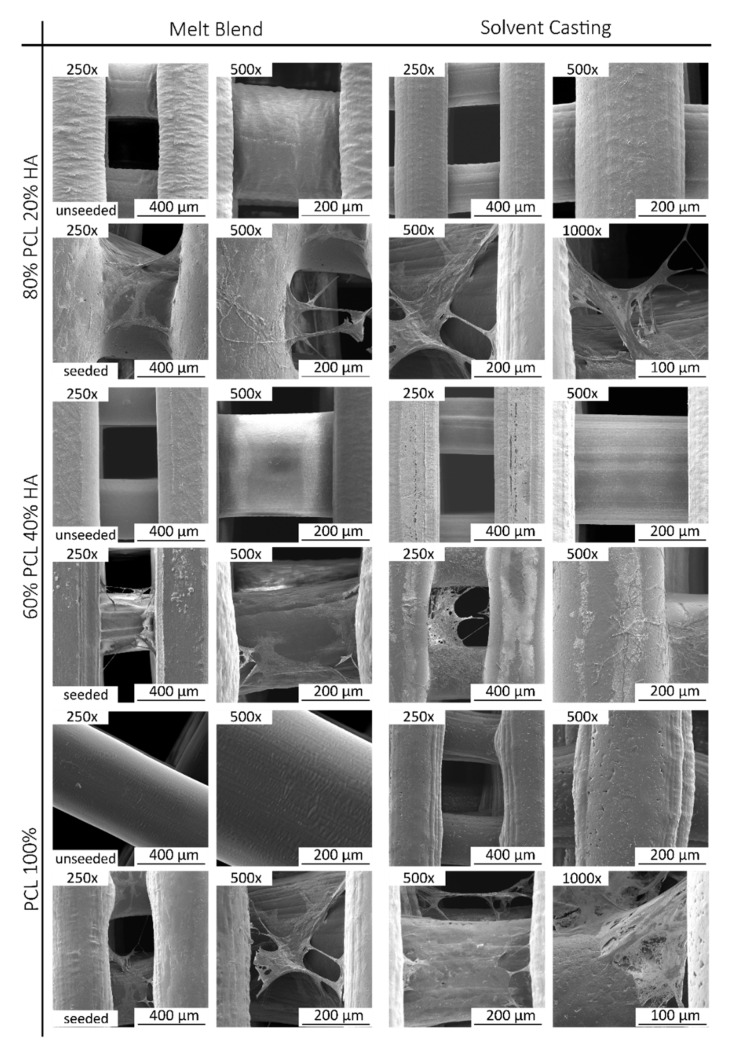
SEM analyses of the seeded and unseeded scaffolds with different magnifications. Images obtained by a high resolution (Schottky) Environmental Scanning Electron Microscope with X-ray microanalysis and Electron Backscattered Diffraction analysis: Quanta 400 FEG ESEM/EDAX Genesis X4M in high vacuum mode.

**Table 1 ijms-23-02318-t001:** Thermal properties of PCL and PCL/HA composites by DSC, TGA and DTG.

HA Content (wt%)	Method	DSC				TGA	DTG
		Tc (°C)	Tm (°C)	ΔHm(J/g)	Xc (%)	M Loss (%)	Td (°C)
0	MB	38.93 ± 0.07	58.43 ± 0.18	56.65 ± 0.29	41.33 ± 0.21	99.46 ± 0.40	386.52 ± 1.85
	SC	38.13 ± 0.11	58.31 ± 0.12	56.48 ± 0.20	40.49± 0.15	99.30 ± 0.58	387.55 ± 0.71
20	MB	38.68 ± 0.15	58.83 ± 0.13	47.24 ± 0.34	42.31 ± 0.22	80.43 ± 0.60	379.37 ± 0.90
	SC	38.26 ± 0.16	58.93 ± 0.54	44.35 ± 0.31	39.74 ± 0.28	79.65 ± 0.19	378.93 ± 1.45
40	MB	36.63 ± 0.27	57.65 ± 0.60	36.02 ± 0.45	43.04 ± 0.54	63.46 ± 3.10	369.19 ± 1.03
	SC	37.08 ± 0.07	58.74 ± 0.29	33.60 ± 0.37	40.14 ± 0.44	60.10 ± 0.33	370.95 ± 0.67

**Table 2 ijms-23-02318-t002:** Mechanical properties of compressive strength of scaffolds.

HA Content (wt%)	Method	Compressive Modulus (MPa)	σ Max (50% Strain) (MPa)
0	MB	58.30 ± 0.78	15.09 ± 2.78
	SC	26.76 ± 1.70	8.38 ± 0.45
20	MB	81.01 ± 1.59	20.05 ± 2.04
	SC	41.59 ± 1.31	12.18 ± 0.76
40	MB	84.84 ±3.26	17.59 ± 2.05
	SC	43.43 ± 1.09	13.93 ± 2.44

**Table 3 ijms-23-02318-t003:** Preliminary cytocompatibility assessment of the PCL groups by Presto Blue^TM^ viability assay for hDPSCs. Corrected absorbance results are presented in mean ± SE.

	PCL MB	PCL SC	hDPSCs Control
24 h	0.065 ± 0.004	0.051 ± 0.003	0.046 ± 0.004
72 h	0.077 ± 0.004	0.078 ± 0.005	0.066 ± 0.005
120 h	0.111 ± 0.002	0.095 ± 0.003	0.086 ± 0.005
168 h	0.131 ± 0.003	0.121 ± 0.005	0.105 ± 0.003

**Table 4 ijms-23-02318-t004:** Cytocompatibility assessed by Presto Blue^TM^ viability assay for hDPSCs. Corrected absorbance results are presented in mean ± SE.

	PCL MB	MB		SC		hDPSCs Control
		PCL/HA 80:20	PCL/HA 60:40	PCL/HA 80:20	PCL/HA 60:40	
24 h	0.060 ± 0.004	0.068 ± 0.004	0.075 ± 0.001	0.069 ± 0.001	0.077 ± 0.002	0.055 ± 0.004
72 h	0.083 ± 0.004	0.098 ± 0.001	0.101 ± 0.002	0.095 ± 0.001	0.113 ± 0.002	0.076 ± 0.005
120 h	0.115 ± 0.003	0.127 ± 0.003	0.131 ± 0.005	0.128 ± 0.003	0.131 ± 0.004	0.110 ± 0.002
168 h	0.122 ± 0.003	0.133 ± 0.003	0.143 ± 0.005	0.134 ± 0.003	0.137 ± 0.004	0.114 ± 0.004

**Table 5 ijms-23-02318-t005:** Cytocompatibility assessed by Presto Blue^TM^ viability assay for hDPSCs. Results of % viability inhibition are presented in mean ± SE, normalized to the PCL MB as 0%.

	MB		SC		hDPSCs Control
	PCL/HA 80:20	PCL/HA 60:40	PCL/HA 80:20	PCL/HA 60:40	
24 h	−15.15 ± 6.09	−26.39 ± 1.14	−16.20 ± 1.04	−30.25 ± 3.56	7.81 ± 6.64
72 h	−18.00 ± 1.38	−21.57 ± 2.51	−15.43 ± 1.35	−36.71 ± 1.96	7.46 ± 5.64
120 h	−9.83 ± 2.73	−13.55 ± 4.47	−11.13 ± 2.29	−13.88 ± 3.28	4.45 ± 1.76
168 h	−8.87 ± 2.05	−17.32 ± 4.09	−9.63 ± 2.15	−12.39 ± 3.02	6.26 ± 2.95

**Table 6 ijms-23-02318-t006:** ARS semi-quantification in mM between groups. The results are presented in mean ± SE. “Undif.” Stands for “Undiferentiated” and “Dif.” Stands for “Differentiated”.

	PCL MB	MB		SC		hDPSCs Control
		PCL/HA 80:20	PCL/HA 60:40	PCL/HA 80:20	PCL/HA 60:40	
Undif.	0.184 ± 0.004	0.422 ± 0.043	0.494 ± 0.015	0.332 ± 0.031	0.500 ± 0.024	0.053 ± 0.004
Dif.	0.803 ± 0.046	1.311 ± 0.098	1.442 ± 0.073	1.145 ± 0.051	1.324 ± 0.069	0.477 ± 0.025

**Table 7 ijms-23-02318-t007:** ARS semi-quantification in mM between groups. The results of % mineral deposition enhancement are presented in mean ± SE, normalized to the PCL MB as 0%.

	MB		SC		hDPSCs Control
	PCL/HA 80:20	PCL/HA 60:40	PCL/HA 80:20	PCL/HA 60:40	
Undif.	129.7 ± 23.03	168.9 ± 8.41	80.88 ± 16.99	172.6 ± 13.25	−70.98 ± 2.307
Dif.	63.33 ± 12.25	79.66 ± 9.126	42.64 ± 6.352	65.04 ± 8.652	−40.60 ± 3.084

**Table 8 ijms-23-02318-t008:** Composition of the prepared mixtures.

Sample	Mixture Method	PCL (wt%)	HA (wt%)
PCL MB	MB	100	-
PCL SC	SC	100	-
PCL/HA 80:20 MB	MB	80	20
PCL/HA 80:20 SC	SC	80	20
PCL/HA 60:40 MB	MB	60	40
PCL/HA 60:40 SC	SC	60	40

## Data Availability

The data that support the findings of this study are available from the corresponding author on request.

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
