# Peer review of "3D Printed Poly(𝜀-caprolactone)/Hydroxyapatite Scaffolds for Bone Tissue Engineering: A Comparative Study on a Composite Preparation by Melt Blending or Solvent Casting Techniques and the Influence of Bioceramic Content on Scaffold Properties"

_ijms, 2022, doi:10.3390/ijms23042318_

Round 1

Reviewer 1 Report

The manuscript entitled ‘3D printed Poly(ε-caprolactone)/Hydroxyapatite scaffolds for Bone Tissue Engineering: A comparative study on composite preparation by melt blending or solvent casting techniques and influence of bioceramic content on scaffold properties’ presents a number of thermoplastics based composite scaffolds produced by two manufacturing methods and by varying the percentages of the matrix (PCL) and reinforcement (hydroxyapatite). The manuscript needs major revision and some parts should be removed, while others should be re-formulated:

  1. The term ‘Device’ refers to a ‘sample or specimen’. The ‘device’ is a thing made or adapted for a particular purpose, especially a piece of mechanical or electronic equipment. The authors referred to a ‘device’ instead of sample/specimen/scaffold etc. This should be accordingly replaced in the whole manuscript.
  2. Does the manufacturing method (Biomate, Line 466) allow large scale reproducibility of scaffolds? In the introduction it is mentioned several times that the purpose of the investigation was to produce synthetic tissue engineering; it is well known that the commercializing of such scaffolds is manufacturing dependent. Consequently, the question arises whether a homemade manufacturing device can provide this facility. A comparative study with scaffolds produced by standardized novel manufacturing devices (3D printing) would have been extremely useful. If tests with animal model and clinical trials for the standardization of the method and the designed scaffolds are not foreseen and this is not the general goal, then it should be mentioned in the manuscript – introduction- that the study is for basic research purposes only and the introduction should be accordingly modified to describe the lab research on similar materials and to present comparation with the state of the art.
  3. An error in several lines (110, 118, 126, 127 etc.) interrupts the continuity of the text with a phrase in Portuguese: Erro! A origem da referência não foi encontrada.
  4. For the affirmation in Line 162: ‘PCL/HA scaffolds present a rougher surface, …’ AFM measurements are needed. Otherwise, it should be eliminated.
  5. Line 255 Figs. 8 and 9 coincide with Figure 9. Cytocompatibility assessed by Presto…or osteogenic differentiation? Where is the Fig. showing Osteogenic differentiation assay…??!!
  6. The ‘normal adhesion’ Line 282, cannot be assessed by observing the SEM images. The adhesion could be quantified by FAK staining. So therefor it shouldn’t be mentioned at all.
  7. The EDX analysis is not discussed in the text and in any case, there is no sense to analyze the material and cells because the system is too complex with many traces, salts, impurities etc. and no realistic interpretation can be given. It is better to eliminate the EDX image.
  8. It is affirmed (Lines 297-298): The conventional techniques include MB and SC, while the new fabrication methods consist of 3D printing technologies, such as extrusion.’ ?! The manuscript presents results on two types of scaffold categories according to the manufacturing method: solvent casting and melt blending. The produced scaffolds are not categorized in the manuscript through conventional vs. new fabrication technique criterion so the affirmation is not sustained through the presented research. The way materials are presented confuse the reader.
  9. Lines 298-299: ‘The optimization of production processes for the dispersion of nanomaterials in polymer matrices…’ There is no suggested method for such an optimization in the manuscript. It should be added in conclusions.
  10. Line 411: ‘…although this assay is not directly described in the guideline, the authors adapted the interpretation from the MTT assay…’ What sort of adaptation? Maybe some correlation has been made? It is better not to mention standards if they are not strictly followed.
  11. Conclusions are poor and should be developed more with respect to the superior material (PLA/HA MB) and the properties of interest

Author Response

Manuscript ID ijms-1577642

Title 3D printed Poly(?-caprolactone)/Hydroxyapatite scaffolds for Bone Tissue Engineering: A comparative study on composite preparation by melt blending or solvent casting techniques and influence of bioceramic content on scaffold properties

Section Materials Science

Special Issue Interactions of Cells with Biomaterials for Regenerative Medicine 2.0

Review Report 1

The manuscript entitled ‘3D printed Poly(ε-caprolactone)/Hydroxyapatite scaffolds for Bone Tissue Engineering: A comparative study on composite preparation by melt blending or solvent casting techniques and influence of bioceramic content on scaffold properties’ presents a number of thermoplastics based composite scaffolds produced by two manufacturing methods and by varying the percentages of the matrix (PCL) and reinforcement (hydroxyapatite). The manuscript needs major revision and some parts should be removed, while others should be re-formulated:

  1. The term ‘Device’ refers to a ‘sample or specimen’. The ‘device’ is a thing made or adapted for a particular purpose, especially a piece of mechanical or electronic equipment. The authors referred to a ‘device’ instead of sample/specimen/scaffold etc. This should be accordingly replaced in the whole manuscript.

The authors acknowledge the Reviewer’s suggestion and accordingly replaced “device” for “scaffold” along the whole manuscript.

  1. Does the manufacturing method (Biomate, Line 466) allow large scale reproducibility of scaffolds? In the introduction it is mentioned several times that the purpose of the investigation was to produce synthetic tissue engineering; it is well known that the commercializing of such scaffolds is manufacturing dependent. Consequently, the question arises whether a homemade manufacturing device can provide this facility. A comparative study with scaffolds produced by standardized novel manufacturing devices (3D printing) would have been extremely useful. If tests with animal model and clinical trials for the standardization of the method and the designed scaffolds are not foreseen and this is not the general goal, then it should be mentioned in the manuscript – introduction- that the study is for basic research purposes only and the introduction should be accordingly modified to describe the lab research on similar materials and to present comparation with the state of the art.

The authors would like to acknowledge the Reviewer’s suggestion. The group has gained expertise in the development of biomanufacturing systems that have been referenced in patents and/or research papers over the last years, named as Bioextruder, Biocell, Dual bioextruder, Biomate, among others.  They allow to produce scaffolds with multiple biomaterials for tissue engineering applications with high reproducibility, comparable with commercialised systems and allowing large scale reproducibility. Further animal model assays are foreseen. With this regard, and considering the Reviewer’s suggestions, the following paragraphs have been modified and updated in the Introduction section:

“PCL, through its melt processing, allows to obtain 3D porous scaffolds with highly interconnected porous network and with high reproducibility by using commercialised [21] and novel biomanufacturing systems [5, 8-12, 22-24]. PCL based scaffolds have been produced by these new developed equipment and have been tested in vitro and in vivo with good cell viability, osteogenic differentiation promotion and improved bone tissue regeneration outcomes, both in vitro and in vivo [5, 8, 11, 25-31].

In this work, in addition to the incorporation of a ceramic amount of 20 and 40 wt.% on PCL, we analysed the influence of the two mixture techniques (solvent casting (SC) and melt blending (MB)) on the properties of PCL and PCL/HA composites. Following, the six different groups considered (PCL SC, PCL MB, PCL/HA 80:20 SC, PCL/HA 80:20 MB, PCL/HA 60:40 SC and PCL/HA 60:40 MB) were 3D printed through an extrusion-based technique, using a previously developed  biomanufacturing system [8-10, 12, 22, 24-29], and their physicochemical properties were further analysed. Following, in vitro assays were taken, applying mesenchymal stem cells, as to evaluate the regenerative potential of the produced scaffolds.

  1. An error in several lines (110, 118, 126, 127 etc.) interrupts the continuity of the text with a phrase in Portuguese: Erro! A origem da referência não foi encontrada.

The authors acknowledge the Reviewer’s comment. The error was probably due to some formatting incompatibilities. Error appeared instead of figures and tables’ references. Manuscript has been carefully revised and corrected references to figures and tables were inserted.

  1. For the affirmation in Line 162: ‘PCL/HA scaffolds present a rougher surface, …’ AFM measurements are needed. Otherwise, it should be eliminated.

The authors acknowledge the Reviewer’s comment. Accordingly, the referred sentence has been eliminated and the following paragraph has been modified in the results section:

“On the other hand, PCL/HA scaffolds present a homogeneous distribution of the HA on the polymer matrix, with some particles exposed on the filament surface.”

  1. Line 255 Figs. 8 and 9 coincide with Figure 9. Cytocompatibility assessed by Presto…or osteogenic differentiation? Where is the Fig. showing Osteogenic differentiation assay…??!!

The authors acknowledge the Reviewer’s comment. Figure 8 was by mistake uploaded twice, replacing figure 9. Figure 9 is now correctly presenting the osteogenic differentiation assay.

  1. The ‘normal adhesion’ Line 282, cannot be assessed by observing the SEM images. The adhesion could be quantified by FAK staining. So therefor it shouldn’t be mentioned at all.

The authors acknowledge the Reviewer’s comment. They only wanted to refer that the cells had the ability to positively attach to the scaffolds’ surface and maintain their adhesion along the duration of the experiment. Nonetheless, and as suggested, “normal adhesion” has been eliminated from the manuscript section.

  1. The EDX analysis is not discussed in the text and in any case, there is no sense to analyze the material and cells because the system is too complex with many traces, salts, impurities etc. and no realistic interpretation can be given. It is better to eliminate the EDX image.

The authors acknowledge the Reviewer’s comment and Figure 11 has been eliminated from the manuscript.

  1. It is affirmed (Lines 297-298): The conventional techniques include MB and SC, while the new fabrication methods consist of 3D printing technologies, such as extrusion.’ ?! The manuscript presents results on two types of scaffold categories according to the manufacturing method: solvent casting and melt blending. The produced scaffolds are not categorized in the manuscript through conventional vs. new fabrication technique criterion so the affirmation is not sustained through the presented research. The way materials are presented confuse the reader.

The authors acknowledge the Reviewer’s comment and modification were accordingly made to the abstract and discussion section of the manuscript.

The manuscript presents results on two types of scaffolds according to the composite manufacturing technique. Melt blending (MB) and solvent casting (SC) are two different fabrication methods used for the dispersion of nanohydroxyapatite in polycaprolactone matrices. Further, the authors applied a non-conventional method, extrusion-based 3D printing, to obtain the 3D scaffolds.

The following paragraph was modified in the abstract section:

“In this study, hydroxyapatite was incorporated on polycaprolactone based scaffolds with two different proportions, 80:20 and 60:40. Scaffolds were produced with two different blending methods, solvent casting and melt blending. The prepared composites were 3D printed through an extrusion-based technique and further investigated with regards to their chemical, thermal, morphological, and mechanical characteristics. In vitro cytocompatibility and osteogenic differentiation was also assessed with human dental pulp stem/stromal cells.”

The following affirmation was removed in the discussion section:

 “The conventional techniques include MB and SC, while the new fabrication methods consist of 3D printing technologies, such as extrusion.”

  1. Lines 298-299: ‘The optimization of production processes for the dispersion of nanomaterials in polymer matrices…’ There is no suggested method for such an optimization in the manuscript. It should be added in conclusions.

The authors acknowledge the Reviewer’s comment, and the referred sentence has been accordingly modified.

The goal of this research paper was to investigate two different production processes already known to disperse nanomaterials in polymer matrices, so no optimization was, in fact, done. The following paragraph was modified in the discussion section and reference to optimization has been eliminated:

“The research on production processes for the dispersion of nanomaterials in polymer matrices and the design of 3D printing scaffolds, as well the combination of these features, play a critical role in tissue engineering.”

  1. Line 411: ‘…although this assay is not directly described in the guideline, the authors adapted the interpretation from the MTT assay…’ What sort of adaptation? Maybe some correlation has been made? It is better not to mention standards if they are not strictly followed.

The authors acknowledge the Reviewer’s comment. The MTT assay is referred as a possible testing method in the guidelines. However, we used the Presto Blue assay. Both are very similar dye-based assays, for cell viability measurement. The latter has been extensively used for this purpose, with comparable performance outcomes to the MTTassay (10.1016/j.tiv.2015.02.003), and allowing live cell-assay, thus, implying the use of less scaffolds and cells (Reduce and Refine). It also allows to evaluate the exact same cell population and scaffold over time, thus degradation of the sample and cell proliferation effect on the results can be more accurate by this method. Interpretation of the results has been further made, following the guidelines criteria (more than 30% inhibition considered cytotoxic). Altogether, both viability assessment methods are very similar, and scientific works have been applying the Presto Blue assay for these purposes with reference to the ISO 10993-5 guideline (10.3390/nano11051152 and 10.1557/mrc.2020.46).

The authors made changes to the paragraph, according to the Reviewer’s suggestion:

“Furthermore, data were analysed according to manufacturing instructions and results were interpreted following ISO 10993-5:2009 “Biological evaluation of medical devices” – Part 5 – “Test for in vitro cytotoxicity” guidelines. The PrestoBlueTM assay was used, as it allows live-cell evaluations [40]. Thus, the same cell population and the same scaffolds can be analysed throughout the duration of the experiment.”

  1. Conclusions are poor and should be developed more with respect to the superior material (PLA/HA MB) and the properties of interest

The authors acknowledge the Reviewer’s comment. The conclusion has been updated accordingly:

“As for the HA incorporation proportion, overall, no statistically significant differences were observed, although, morphologically, the 60:40 presented the best homogeneity requirements.  Considering these preliminary observations, the PCL/HA MB scaffolds presented overall the best outcomes, regarding their mechanical characterization, as well as the in vitro cytocompatibility and osteogenic-promoting potential. They can thus, represent fair candidates for bone tissue engineering studies and further in vitro and in vivo studies are envisioned as to reinforce and support these findings.”

Reviewer 2 Report

  1. The authors used same figures, results in Figure 8 and Figure 9. Thus, the reviewer cannot completely review this manuscript.

  1. The concept of this research is well known and similar research has been already published by other group (A. G. Garcia et al. ACS Biomater. Sci. Eng., 2018, 4, 3317-3326). The authors should mention the novelty and/or strong point of your research compared with previous case in discussion part.

  1. In figure 1B, the authors showed EDX spectrum and suggested HA is detected as distributed throughout the filaments of composites. From only spectra, the reviewers can understand HA is detected from the filaments but cannot understand whether HA is distributed homogeneously or not. The authors should show the Ca, P elemental mapping image of EDX/SEM.

  1. In figure 1B, the authors also showed Ca atomic% and P atomic% of the composite. These elements are derived from only HA and general Ca/P ratio of HA is 1.67. The author’s results are not suitable, and the values are not constant. The authors must discuss this point.

  1. In table 3, 4, 5, 6, 7, the authors said the results are presented in Mean ± SE. However, they showed each mean and SE separately. The reviewer recommends the authors should write these tables like table 1.

  1. It is well known that alizarin red also stained hydroxyapatite. In this case, the authors mixed reagent grade hydroxyapatite with PCL and it is easy to expect this hydroxyapatite is also stained by alizarin red. The authors should show the picture’s, which is the sample after calcification and staining by alizarin red. The authors should also show the control, the sample (probably PCL/HA 60:40) stained by alizarin red without cell culturing.

Author Response

Manuscript ID ijms-1577642

Title 3D printed Poly(?-caprolactone)/Hydroxyapatite scaffolds for Bone Tissue Engineering: A comparative study on composite preparation by melt blending or solvent casting techniques and influence of bioceramic content on scaffold properties

Section Materials Science

Special Issue Interactions of Cells with Biomaterials for Regenerative Medicine 2.0

Review Report 1

The manuscript entitled ‘3D printed Poly(ε-caprolactone)/Hydroxyapatite scaffolds for Bone Tissue Engineering: A comparative study on composite preparation by melt blending or solvent casting techniques and influence of bioceramic content on scaffold properties’ presents a number of thermoplastics based composite scaffolds produced by two manufacturing methods and by varying the percentages of the matrix (PCL) and reinforcement (hydroxyapatite). The manuscript needs major revision and some parts should be removed, while others should be re-formulated:

  1. The term ‘Device’ refers to a ‘sample or specimen’. The ‘device’ is a thing made or adapted for a particular purpose, especially a piece of mechanical or electronic equipment. The authors referred to a ‘device’ instead of sample/specimen/scaffold etc. This should be accordingly replaced in the whole manuscript.

The authors acknowledge the Reviewer’s suggestion and accordingly replaced “device” for “scaffold” along the whole manuscript.

  1. Does the manufacturing method (Biomate, Line 466) allow large scale reproducibility of scaffolds? In the introduction it is mentioned several times that the purpose of the investigation was to produce synthetic tissue engineering; it is well known that the commercializing of such scaffolds is manufacturing dependent. Consequently, the question arises whether a homemade manufacturing device can provide this facility. A comparative study with scaffolds produced by standardized novel manufacturing devices (3D printing) would have been extremely useful. If tests with animal model and clinical trials for the standardization of the method and the designed scaffolds are not foreseen and this is not the general goal, then it should be mentioned in the manuscript – introduction- that the study is for basic research purposes only and the introduction should be accordingly modified to describe the lab research on similar materials and to present comparation with the state of the art.

The authors would like to acknowledge the Reviewer’s suggestion. The group has gained expertise in the development of biomanufacturing systems that have been referenced in patents and/or research papers over the last years, named as Bioextruder, Biocell, Dual bioextruder, Biomate, among others.  They allow to produce scaffolds with multiple biomaterials for tissue engineering applications with high reproducibility, comparable with commercialised systems and allowing large scale reproducibility. Further animal model assays are foreseen. With this regard, and considering the Reviewer’s suggestions, the following paragraphs have been modified and updated in the Introduction section:

“PCL, through its melt processing, allows to obtain 3D porous scaffolds with highly interconnected porous network and with high reproducibility by using commercialised [21] and novel biomanufacturing systems [5, 8-12, 22-24]. PCL based scaffolds have been produced by these new developed equipment and have been tested in vitro and in vivo with good cell viability, osteogenic differentiation promotion and improved bone tissue regeneration outcomes, both in vitro and in vivo [5, 8, 11, 25-31].

In this work, in addition to the incorporation of a ceramic amount of 20 and 40 wt.% on PCL, we analysed the influence of the two mixture techniques (solvent casting (SC) and melt blending (MB)) on the properties of PCL and PCL/HA composites. Following, the six different groups considered (PCL SC, PCL MB, PCL/HA 80:20 SC, PCL/HA 80:20 MB, PCL/HA 60:40 SC and PCL/HA 60:40 MB) were 3D printed through an extrusion-based technique, using a previously developed  biomanufacturing system [8-10, 12, 22, 24-29], and their physicochemical properties were further analysed. Following, in vitro assays were taken, applying mesenchymal stem cells, as to evaluate the regenerative potential of the produced scaffolds.

  1. An error in several lines (110, 118, 126, 127 etc.) interrupts the continuity of the text with a phrase in Portuguese: Erro! A origem da referência não foi encontrada.

The authors acknowledge the Reviewer’s comment. The error was probably due to some formatting incompatibilities. Error appeared instead of figures and tables’ references. Manuscript has been carefully revised and corrected references to figures and tables were inserted.

  1. For the affirmation in Line 162: ‘PCL/HA scaffolds present a rougher surface, …’ AFM measurements are needed. Otherwise, it should be eliminated.

The authors acknowledge the Reviewer’s comment. Accordingly, the referred sentence has been eliminated and the following paragraph has been modified in the results section:

“On the other hand, PCL/HA scaffolds present a homogeneous distribution of the HA on the polymer matrix, with some particles exposed on the filament surface.”

  1. Line 255 Figs. 8 and 9 coincide with Figure 9. Cytocompatibility assessed by Presto…or osteogenic differentiation? Where is the Fig. showing Osteogenic differentiation assay…??!!

The authors acknowledge the Reviewer’s comment. Figure 8 was by mistake uploaded twice, replacing figure 9. Figure 9 is now correctly presenting the osteogenic differentiation assay.

  1. The ‘normal adhesion’ Line 282, cannot be assessed by observing the SEM images. The adhesion could be quantified by FAK staining. So therefor it shouldn’t be mentioned at all.

The authors acknowledge the Reviewer’s comment. They only wanted to refer that the cells had the ability to positively attach to the scaffolds’ surface and maintain their adhesion along the duration of the experiment. Nonetheless, and as suggested, “normal adhesion” has been eliminated from the manuscript section.

  1. The EDX analysis is not discussed in the text and in any case, there is no sense to analyze the material and cells because the system is too complex with many traces, salts, impurities etc. and no realistic interpretation can be given. It is better to eliminate the EDX image.

The authors acknowledge the Reviewer’s comment and Figure 11 has been eliminated from the manuscript.

  1. It is affirmed (Lines 297-298): The conventional techniques include MB and SC, while the new fabrication methods consist of 3D printing technologies, such as extrusion.’ ?! The manuscript presents results on two types of scaffold categories according to the manufacturing method: solvent casting and melt blending. The produced scaffolds are not categorized in the manuscript through conventional vs. new fabrication technique criterion so the affirmation is not sustained through the presented research. The way materials are presented confuse the reader.

The authors acknowledge the Reviewer’s comment and modification were accordingly made to the abstract and discussion section of the manuscript.

The manuscript presents results on two types of scaffolds according to the composite manufacturing technique. Melt blending (MB) and solvent casting (SC) are two different fabrication methods used for the dispersion of nanohydroxyapatite in polycaprolactone matrices. Further, the authors applied a non-conventional method, extrusion-based 3D printing, to obtain the 3D scaffolds.

The following paragraph was modified in the abstract section:

“In this study, hydroxyapatite was incorporated on polycaprolactone based scaffolds with two different proportions, 80:20 and 60:40. Scaffolds were produced with two different blending methods, solvent casting and melt blending. The prepared composites were 3D printed through an extrusion-based technique and further investigated with regards to their chemical, thermal, morphological, and mechanical characteristics. In vitro cytocompatibility and osteogenic differentiation was also assessed with human dental pulp stem/stromal cells.”

The following affirmation was removed in the discussion section:

 “The conventional techniques include MB and SC, while the new fabrication methods consist of 3D printing technologies, such as extrusion.”

  1. Lines 298-299: ‘The optimization of production processes for the dispersion of nanomaterials in polymer matrices…’ There is no suggested method for such an optimization in the manuscript. It should be added in conclusions.

The authors acknowledge the Reviewer’s comment, and the referred sentence has been accordingly modified.

The goal of this research paper was to investigate two different production processes already known to disperse nanomaterials in polymer matrices, so no optimization was, in fact, done. The following paragraph was modified in the discussion section and reference to optimization has been eliminated:

“The research on production processes for the dispersion of nanomaterials in polymer matrices and the design of 3D printing scaffolds, as well the combination of these features, play a critical role in tissue engineering.”

  1. Line 411: ‘…although this assay is not directly described in the guideline, the authors adapted the interpretation from the MTT assay…’ What sort of adaptation? Maybe some correlation has been made? It is better not to mention standards if they are not strictly followed.

The authors acknowledge the Reviewer’s comment. The MTT assay is referred as a possible testing method in the guidelines. However, we used the Presto Blue assay. Both are very similar dye-based assays, for cell viability measurement. The latter has been extensively used for this purpose, with comparable performance outcomes to the MTTassay (10.1016/j.tiv.2015.02.003), and allowing live cell-assay, thus, implying the use of less scaffolds and cells (Reduce and Refine). It also allows to evaluate the exact same cell population and scaffold over time, thus degradation of the sample and cell proliferation effect on the results can be more accurate by this method. Interpretation of the results has been further made, following the guidelines criteria (more than 30% inhibition considered cytotoxic). Altogether, both viability assessment methods are very similar, and scientific works have been applying the Presto Blue assay for these purposes with reference to the ISO 10993-5 guideline (10.3390/nano11051152 and 10.1557/mrc.2020.46).

The authors made changes to the paragraph, according to the Reviewer’s suggestion:

“Furthermore, data were analysed according to manufacturing instructions and results were interpreted following ISO 10993-5:2009 “Biological evaluation of medical devices” – Part 5 – “Test for in vitro cytotoxicity” guidelines. The PrestoBlueTM assay was used, as it allows live-cell evaluations [40]. Thus, the same cell population and the same scaffolds can be analysed throughout the duration of the experiment.”

  1. Conclusions are poor and should be developed more with respect to the superior material (PLA/HA MB) and the properties of interest

The authors acknowledge the Reviewer’s comment. The conclusion has been updated accordingly:

“As for the HA incorporation proportion, overall, no statistically significant differences were observed, although, morphologically, the 60:40 presented the best homogeneity requirements.  Considering these preliminary observations, the PCL/HA MB scaffolds presented overall the best outcomes, regarding their mechanical characterization, as well as the in vitro cytocompatibility and osteogenic-promoting potential. They can thus, represent fair candidates for bone tissue engineering studies and further in vitro and in vivo studies are envisioned as to reinforce and support these findings.”

Review Report 2

  1. The authors used same figures, results in Figure 8 and Figure 9. Thus, the reviewer cannot completely review this manuscript.

The authors acknowledge the Reviewer’s comment. Figure 8 was by mistake uploaded twice, replacing figure 9. Figure 9 is now correctly presenting the osteogenic differentiation assay.

  1. The concept of this research is well known and similar research has been already published by other group (A. G. Garcia et al. ACS Biomater. Sci. Eng., 2018, 4, 3317-3326). The authors should mention the novelty and/or strong point of your research compared with previous case in discussion part.

The authors acknowledge the Reviewer’s comment. This work intended to analyse the impact of different proportions of HA and composite fabrication methods on the mechanical, physical, chemical, and in vitro outcomes of 3D scaffolds. In this work, the authors compare different blending methods (MB and SC) while Garcia and colleagues only used SC method. Beyond the aim of this study to systematically compare two fabrication processes for the incorporation of HA nanoparticles in PCL matrices, we explore the impact of different HA ratio (60:40 and 80:20) on the physicochemical characterization and biological functions of 3D scaffolds, while in the referred work only one was assessed (10% wt). Furthermore, the HA was melted with the PCL prior to the extrusion in the case of this work, while Garcia and colleagues electrosprayed layers of PCL and HA separately. The extrusion technique used in the present work presents some advantages comparing with electrospinning method, as it allows to fabricate scaffolds with suitable mechanical properties, controlled pore sizes, and well-interconnected pores, which are some of the limitations of scaffolds obtained by electrospinning.

The authors believe this study to present a valuable contribution to the field, as current challenges in 3D printing composite biomaterials are more focused on the most suitable equilibrium between composite materials and 3D printing technologies to control scaffold´s morphology and structure, as to obtain 3D structures with adequate properties for Tissue Engineering applications. Moreover, it is well known that more studies are needed to evaluate 3D composite scaffolds regarding their physicochemical and biological characteristics, and to correlate these results and more accurately understand how the distribution of nanoparticles in the polymer matrix and 3D printing technique could influence final scaffold properties. As suggested by the reviewer, references to the mentioned work have been inserted and the following paragraph was included in the discussion section:

“Following previous valuable works on PCL/HA scaffolds with promising outcomes [13, 44, 46-49], this work intended to further analyse the impact of different proportions of HA content, as well as the application of two different fabrication processes for the incorporation of HA nanoparticles in PCL matrices, on the mechanical, physical, chemical, and in vitro outcomes of the scaffolds. Moreover, it is also important to highlight the aim to achieve control, reproducible and well-defined 3D structures fabricated by a previously developed 3D printing system [8-10, 12, 22, 24-29].”

  1. In figure 1B, the authors showed EDX spectrum and suggested HA is detected as distributed throughout the filaments of composites. From only spectra, the reviewers can understand HA is detected from the filaments but cannot understand whether HA is distributed homogeneously or not. The authors should show the Ca, P elemental mapping image of EDX/SEM.

The authors acknowledge the Reviewer’s comment. Figure 1B has been modified, according to the Reviewer’s suggestions.

  1. In figure 1B, the authors also showed Ca atomic% and P atomic% of the composite. These elements are derived from only HA and general Ca/P ratio of HA is 1.67. The author’s results are not suitable, and the values are not constant. The authors must discuss this point.

The authors acknowledge the Reviewer’s comment. As suggested, the following paragraph was included in the discussion section:

“The Ca/P ratios of the scaffolds are also presented in the Figure 1B. Results show that Ca/P ratio is different among the produced scaffolds. The samples produced by MB present Ca/P ratio closer to the stochiometric HA (1.67) [48]. The authors hypothesise these differences to be related with the blending processes, as well as to the presence of some HA particles exposed on the filament surface. These results also support the higher mechanical properties obtained in the samples produced by MB.”

  1. In table 3, 4, 5, 6, 7, the authors said the results are presented in Mean ± SE. However, they showed each mean and SE separately. The reviewer recommends the authors should write these tables like table 1.

The authors acknowledge the Reviewer’s comment. Tables 3 to 7 have been modified, according to the Reviewer’s suggestions.

  1. It is well known that alizarin red also stained hydroxyapatite. In this case, the authors mixed reagent grade hydroxyapatite with PCL and it is easy to expect this hydroxyapatite is also stained by alizarin red. The authors should show the picture’s, which is the sample after calcification and staining by alizarin red. The authors should also show the control, the sample (probably PCL/HA 60:40) stained by alizarin red without cell culturing.

The authors acknowledge the Reviewer’s comment. Although some dye can stain the HA, the osteogenic extension was not evaluated qualitatively. It was, instead, quantitatively assessed using a protocol (https://www.sciencellonline.com/PS/8678.pdf) that entailed the following more important steps: after dying, wells were washed extensively with PBS (5 times) and cells were further fixated with 4% formaldehyde. Images were not taken, as the nature of the devices is not transparent, and the MOI would not capture any valid image. Further, the dye was extracted from the cells. For this purpose, cells were extracted with a cell scraper from the surfaces of the devices and further transferred to a 10% acetic acid solution. By using these steps, the authors only quantify the dye incorporated by the cells, and not by the device itself. Similar works have applied this method in scaffolds containing HA (https://doi.org/10.3390/polym13020257, https://doi.org/10.1002/jbm.a.36388). Normal photographic images were not taken, as the scrapping of the samples would alter their appearance. Also, the authors considered them to induce error, as the staining of the scaffold itself could influence the extension of the matrix staining. However, considering the Reviewers’ comment, the authors will include photographic images of upcoming assays, as well as consider RT-PCR.

Round 2

Reviewer 1 Report

Most of the requirements have been fulfilled.

I suggest publication. 

Reviewer 2 Report

The authors revised to reviewer's comments well and the revised manuscript will be judged to be  worthy of acceptance.